# Online Tracking of Maneuvering Target Trajectory Based on Chaotic Time Series Prediction

**DOI:** 10.3390/e24111668

**Published:** 2022-11-15

**Authors:** Qian Wei, Peng Su, Lin Zhou, Wentao Shi

**Affiliations:** 1School of Artificial Intelligence, Henan University, Zhengzhou 450046, China; 2School of Marine Science and Technology, Northwestern Polytechnical University, Xi’an 710072, China

**Keywords:** online prediction, trajectory segmentation, PSO model identification, feedback optimization, initial value sensitivity

## Abstract

Online prediction of maneuvering target trajectory is one of the most popular research directions at present. Specifically, the primary factors balancing, between prediction accuracy and response time, will give the research substance. This paper presents an online trajectory prediction algorithm based on small sample chaotic time series (OTP-SSCT). First, we optimize in terms of data breadth. The dynamic split window is built according to the motion characteristics of the maneuvering target, thus realizing trajectory segmentation and constructing a small sample chaotic time series prediction set. Second, since fully considering the motion patterns of maneuvering targets, we introduce the spatiotemporal features into the particle swarm optimization (PSO) model identification algorithm, which improves the identification sensitivity of key trajectory data points. Furthermore, we propose a feedback optimization strategy of residual compensation to correct the trajectory prediction values to improve the prediction accuracy. For the initial value sensitivity problem of the PSO model identification algorithm, we propose a new initial population strategy, which improves the effectiveness of initial parameters on model identification. Through simulation experiment analysis, it is verified that the proposed OTP-SSCT algorithm achieves better prediction accuracy and faster response time.

## 1. Introduction

Trajectory prediction combines trajectory information as time series, and uses prediction algorithms to reasonably predict the future movement trend of maneuvering target. It is widely used in unmanned aerial vehicle (UAV) trajectory planning, intelligent transportation system, pedestrian trajectory tracking and other fields [1,2,3]. In recent years, taking advantage of sensor network technical development, the located target has higher mobility and is nonlinear and time-varying. The classical trajectory prediction algorithm with a pre-set motion model usually cannot dynamically adjust the model parameters. Hence, the phenomenon of model mismatch, which leads to inaccurate results and occurs in the trajectory prediction process of strong maneuverability targets. In order to solve the insufficient dynamics problem, many scholars began to study online trajectory prediction methods based on dynamic learning; however, those methods need to consider both prediction accuracy and prediction response time. So, the establishment has processed ultra-big data and the high-speed calculation capacity prediction system—the key question is what kind of method is appropriate.

Generally, online trajectory prediction methods are mainly divided into two main categories. One is the prediction algorithm with pre-set motion model. These methods mainly establish the motion model in advance through the laws of dynamics or physics, and then adjust the model parameters according to the motion state to achieve an online prediction of maneuvering targets [4]. Zhang et al. proposed an online four-dimensional trajectory prediction method [5]. This method improved the accuracy of trajectory prediction by updating the flight intention with the horizontal and vertical intention models of the vehicle, while incorporating the current state. A multimodel-based extended Kalman filter (EKF) was proposed in [6]. The method combined the EKF with the interacting multiple model (IMM) framework to predict the vehicle trajectory by selecting different EKF models. IMM is an adaptive algorithm that can effectively adjust the probability of each model, and is especially suitable for the localization tracking of maneuvering targets [7]. Wu et al. proposed an adaptive trajectory prediction algorithm [8]. By integrating two motion models and adjusting the model parameters adaptively, the algorithm could accurately predict the trajectory of pedestrians. Xie et al. proposed a vehicle trajectory prediction algorithm, which considered both dynamic motion with physical laws in the short term and advanced driving patterns with maneuverability estimation in the long term [9]; thus, the long-term prediction of vehicle trajectories was achieved.

The above methods are suited for maneuvering targets with low motion complexity. For moving targets with high maneuverability, it is difficult to directly establish a prediction model to adapt to the nonlinearity shifts in the track process. For this reason, the complexity of the model will increase sharply, and the robustness of the model will deteriorate. So, the prediction algorithm with a parser motion model can hardly provide good performance for high maneuverability targets.

The other category is data-driven trajectory prediction algorithms, which are divided into two methods, classical prediction methods based on regression theory and model identification methods using artificial intelligence data mining [10,11]. Regression prediction methods use a large number of trajectory data to build regression equations and achieve online prediction through iterative regression [12]. These algorithms include linear regression, logistic regression, ridge regression, autoregressive moving average, etc. Model identification methods focus on mining trajectory information deeply and identify target tracking models by intelligent algorithms; therefore, these methods are enable online trajectory prediction with high trajectory complexity. Liu et al. proposed a trajectory estimation framework for multi-UAV based on the prediction of user’s movement information [13]. The method trains the parameters of the echo state network model by the true trajectory data set, thereby establishing a tracking model to predict the trajectory of the UAV. Shi et al. proposed a flight trajectory prediction model based on a long short-term memory (LSTM) network [14]. The sliding window method was introduced to avoid the dynamic dependence of adjacent states in long-term sequence, which improves the accuracy of online prediction. Ma et al. proposed a deep learning-based four-dimensional trajectory prediction algorithm, in which the flight trajectory features in the spatial and temporal dimensions were extracted by convolutional neural networks and LSTM, respectively [15]. Han et al. proposed a short-term online trajectory prediction model based on the gated cyclic neural network, which was trained and updated with model parameters by the batch method to achieve an online prediction of short-term flight trajectories [16].

Generally, unlike the prediction algorithm of pre-set motion model, the above methods learn the target motion law by mining the trajectory information, and thus achieve online trajectory prediction. By avoiding the physical modeling problem of complex motion, such algorithms are able to track highly maneuverable targets with surprising accuracy. However, the accuracy of the model identification is often related to the amount of training data. Such algorithms need the support of large-scale computation and real time prediction; therefore, processing the data effectively and establishing an accurate model is the bridge between the two aspects.

Due to the highly nonlinear and time-varying characteristics of maneuver target trajectory, a simple time series analysis is difficult to use to analyze the comprehensive reflect target’s motion laws. In recent years, the prediction of chaotic time series has become a popular research direction regarding the aspect of trajectory prediction [17,18]. To analyze the deep behavioral characteristics of maneuvering targets, many methods perform prediction model identification on the basis of chaos theory. Liu et al. proposed a ship trajectory prediction model, which enhanced the optimization effect of the differential evolution algorithm on the model parameters by introducing chaos theory [19]. Hong et al. proposed a motion prediction model of floating platform, which optimized the model parameters by a chaotic efficient bat algorithm, thus achieving complex nonlinear trajectory prediction [20]. Recently, the Volterra series model, as a common model, has been used to predict chaotic time series, which can fit nonlinear data for accurate prediction [21]. Han et al. proposed a local Volterra prediction algorithm based on the clustering of phase points [22]. The method compared the similarity of predicted phase points and observation points in the clustering, and the optimal adjacent phase points were used to train the prediction model. Qiao et al. proposed a novel hybrid prediction model based on the Volterra filter [23]. The model parameters were optimized by an improved whale optimization algorithm. Lv et al. proposed a hybrid dynamic prediction algorithm based on the Volterra prediction model [24]. By introducing the sliding window method, the algorithm realized the online identification of the Volterra model parameters, and the dynamic relations were constructed from the high-dimensional phase space to realize the trajectory prediction of complex systems. Therefore, combining the prediction method with the chaos theory method, the algorithm achieves a better trajectory prediction effect; however, most algorithms based on chaos theory are only suitable for off-line trajectory prediction. For the complex motion model, how to extract the trajectory features in real time and build the prediction model dynamically are still two major problems to be solved. At present, the online trajectory prediction for chaotic time series has some deficiency in dynamic construction of models.

To summarize: in order to achieve stable online prediction of maneuvering target trajectories, this paper proposes an online trajectory prediction algorithm based on small sample chaotic time series (OTP-SSCT). Firstly, a small sample chaotic time series prediction set is constructed using a dynamic sliding window. Secondly, combining the spatiotemporal characteristics of maneuvering targets, the online trajectory prediction model is constructed using the improved PSO algorithm. Finally, the feedback optimization strategy is adopted to further improve the accuracy of the online trajectory prediction. The main contributions of this paper are as follows:

(1) In this paper, an online trajectory prediction method of maneuvering targets is proposed according to chaotic time series analysis. Thus, considering the motion characteristics of maneuvering targets, we construct a small sample chaotic time series prediction set by an innovative trajectory segmentation method. It realizes dynamic segmentation for trajectory data. Since fully considering nonlinear maneuver characteristics of trajectory data, spatiotemporal features are introduced in the improved PSO algorithm to increase the identification sensitivity of key trajectory data points. For the superposition of prediction errors caused by model identification, the trajectory predicted values are corrected by the feedback optimization strategy to ensure the prediction accuracy.

(2) For the initial value sensitivity problem of PSO model identification algorithm, a new initial population strategy is proposed in this paper. According to the spatial distribution of candidate populations, the strategy selects some improvement positions to compensate the uniformity of the initial population distribution, which avoids the algorithm from falling into local traps.

The content of this paper is organized as follows. Section 2 introduces the Volterra prediction model based on phase space reconstruction of chaotic time series. Section 3 elaborates principle and implementation process of the proposed OTP-SSCT algorithm. Further, a new initial population strategy of the PSO model identification algorithm is presented in Section 4. Section 5 gives the pseudo-code of our algorithm. Section 6 verifies the feasibility and effectiveness of the proposed algorithm. Section 7 provides the conclusions of this paper.

## 2. Volterra Prediction Model Based on Phase Space Reconstruction

The trajectory of maneuvering target is highly nonlinear and time-varying, which increases the difficulty of the prediction model identification. In this paper, an online prediction method of maneuver trajectory is proposed according to chaotic time series analysis. In order to obtain more motion laws of the target, the trajectory data can be reconstructed into the phase space by chaos theory [25]. It is easier to fit the evolution law between phase points by the nonlinear prediction model in the high-dimensional phase space. Thus, accurate prediction of maneuvering targets can be achieved. The trajectory prediction process of Volterra model is shown in Figure 1.

Assuming that the observed trajectory one-dimensional time series is $x\left(n\right)$, n=1,2,⋯,N. According to Takens embedding theorem, the phase space reconstruction of the original sequence does not change the evolutionary information of the system [26]. The reconstructed multidimensional time series is represented by phase points as follows:(1)$$X_{l}={\left[x\left(l\right),x\left(l+\tau \right),\cdots ,x\left(l+\left(m-1\right)\tau \right)\right]}^{T}\;l=1,2,\cdots ,L$$
where $X_{l}$ is the *l*-th phase point in phase space. *m* is the embedding dimensions. τ is the delay time. $L=N-\left(m-1\right)\tau$ is the number of phase points. *N* is the number of trajectory data.

In the phase space, there is a mapping $F:R^{m}\to R^{m}$ to represent the evolution law between phase points. Thus, there exists a new mapping $f:R^{m}\to R$ to represent the evolutionary law between trajectory data and phase points, as shown below:(2)$$x\left(l+\left(m-1\right)\tau +1\right)=f\left(X_{l}\right)\;l=1,2,\cdots ,L$$
where $x\left(l+\left(m-1\right)\tau +1\right)$ represents the *m*-th dimensional component of $X_{l+1}$.

The Volterra model can characterize arbitrary nonlinear functions and has good scalability. In this paper, the Volterra model is chosen as the mapping relationship *f* in Equation (2). The *p*-order discrete Volterra model with memory length *m* is expressed as follows:(3)$$y\left(n\right)=\sum_{k=1}^{p}\sum_{i_{1},\cdots ,i_{k}=0}^{m-1}h_{k}\left(i_{1},i_{2},\cdots ,i_{k}\right)\prod_{j=1}^{k}x\left(n-i_{j}\right)\;n=1,2,\cdots ,N$$
where $x\left(n\right)$ and $y\left(n\right)$ is the input and the output, respectively. *p* is the order of Volterra prediction model. *m* is the embedding dimensions (memory length). $h_{k}\left(i_{1},i_{2},\cdots ,i_{k}\right)$ is the *k*-th kernel parameter value, which can be expressed in the form of kernel vector H = [$h_{1}\left(0\right),$$h_{1}\left(1\right),$⋯,$h_{1}\left(m-1\right),$$h_{2}\left(0,0\right),$$h_{2}\left(0,1\right),$⋯,$h_{2}\left(0,m-1\right),$⋯,$h_{p}\left(m-1,m-1\right)$].

Combining Equations (2) and (3), the prediction values $\hat{x}\left(n+1\right)$ are obtained from the Volterra prediction model as follows:(4)$$\hat{x}\left(n+1\right)=\sum_{k=1}^{p}\sum_{i_{1},\cdots ,i_{k}=0}^{m-1}h_{k}\left(i_{1},i_{2},\cdots ,i_{k}\right)\prod_{j=1}^{k}x\left(n-i_{j}\times \tau \right)\;n=1,2,\cdots ,N$$

Determining the kernel parameters value of the Volterra prediction model is a typical nonlinear system identification problem. The OTP-SSCT algorithm is proposed in Section 3. By constructing a small sample chaotic time series prediction set, the parameters of Volterra model are dynamically identified. Thus, online prediction of maneuvering target trajectories is achieved.

## 3. Online Trajectory Prediction Algorithm Based on Small Sample Chaotic Time Series

Traditional prediction algorithms with pre-set motion model will rarely change the model parameters, resulting in insufficient dynamic capabilities. In contrast, data-fitting based prediction algorithms require large-scale training data support, resulting in poor online performance. To overcome these problems, this paper proposes the OTP-SSCT algorithm, which can dynamically construct the prediction model and realize the online prediction of maneuvering trajectory.

### 3.1. Algorithm Model Framework

The framework diagram of the proposed algorithm is shown in Figure 2. To begin with, the input trajectory data are pre-processed by the presented trajectory segmentation method. The dynamic split window is built according to the motion characteristics of maneuvering targets, thus realizing trajectory segmentation and constructing a small sample chaotic time series prediction set. Secondly, in combination with that prediction set, we construct an online trajectory prediction model through improving the PSO model identification algorithm. In addition, we propose an initial population strategy to improve the effectiveness of initial parameters on model identification. Finally, while outputting the trajectory prediction values, we also compensate them by the feedback optimization strategy, so as to improve the prediction accuracy.

### 3.2. Dynamic Trajectory Segmentation

The long-term movement of the maneuvering target generates a large amount of redundant historical trajectory information, some of which is no longer valid or even interferes with the construction of the prediction model. In this section, we propose an innovative trajectory segmentation method, which is used to construct a small sample chaotic time series prediction set of the target. The method achieves trajectory segmentation through the dynamic split window. Firstly, we design a similarity function to measure the dynamic changes of the trajectory data, and also define the magnitude of the similarity value by the threshold method. Secondly, by extracting the motion characteristics of the maneuvering target, we give the adjustment rules for the length of dynamic split window. Finally, the trajectory segmentation is performed through the dynamic split window, and a small sample chaotic time series prediction set is constructed for the subsequent model identification. The flow chart of trajectory segmentation with dynamic split window is shown in Figure 3.

As the target trajectory information updates, redundant historical trajectories need to be eliminated. The dynamic split window focuses on the latest trajectory information of the target. To accomplish this goal, the dynamic time warping (DTW) distance is used as the similarity function [27]. When the similarity value is greater than the preset threshold $TH_{dtw}$ ($TH_{dtw}$ is determined by the actual data), it indicates that the target produces a new maneuver. At this moment, it is necessary to adjust the length of dynamic split window so that the segmentation trajectory discards useless information. After each length adjustment, we save the first dynamic split window as $SW_{old}$, while the latest dynamic split window is $SW_{new}$. The similarity function is as follows:(5)$$DTW\left(SW_{new},SW_{old}\right)=\underset{P=p_{1},\cdots ,p_{k},\cdots ,p_{K}}{\arg\min}\sqrt{\sum_{k=1}^{K}{\left[p_{k}\left(i,j\right)\right]}^{2}}$$
where
$$p_{k}\left(i,j\right)={\left(x(i)-x(j)\right)}^{2}+\min\left\{p_{k}\left(i-1,j-1\right),p_{k}\left(i-1,j\right),p_{k}\left(i,j-1\right)\right\}$$
*P* is the warping path between $SW_{new}$ and $SW_{old}$. $K\in \left[W,2W-1\right)$ is the length of the warping path. *W* is the length of dynamic split window. $x(i)\in SW_{new}$ and $x(j)\in SW_{old}$ are the trajectory data, 1≤i,j≤W. More details can be found in [27].

The length of the dynamic split window is determined by the motion characteristics of the maneuvering target. At first, we define that the motion characteristics consist of two components, trajectory data mean volatility ratio $R_{V}$ and tracking model error volatility ratio $R_{E}$. The calculation methods are shown below, respectively:(6)$$R_{V}=\frac{V_{new}-V_{old}}{V_{old}}$$
(7)$$R_{E}=\frac{E_{new}-E_{old}}{E_{old}}$$
where
(8)$$V_{new}=\frac{\mathrm{Var}\left(SW_{new}\right)}{\mathrm{Max}\left(SW_{new}\right)-\mathrm{Min}\left(SW_{new}\right)}$$
(9)$$E_{new}=\frac{1}{W}\sum_{l=1}^{W}\left|x\left(l\right)-\hat{x}\left(l\right)\right|\;x\left(l\right)\in SW_{new}$$
$V_{old}$ and $E_{old}$ are calculated in the same way as Equations (8) and (9). $V_{new}$ and $V_{old}$ are the fluctuation values of $SW_{new}$ and $SW_{old}$, respectively, indicating the intensity of the target motion. Var denotes calculation of variance. Max and Min denote calculation of the maximum and minimum values, respectively. $E_{new}$ and $E_{old}$ are the average prediction errors of $SW_{new}$ and $SW_{old}$, respectively, indicating the accuracy of the prediction model. $\hat{x}\left(l\right)$ and $x\left(l\right)$ denote the prediction value and true value of the trajectory, respectively.

The dynamic split window length *W* can be dynamically adjusted according to the motion characteristics of Equations (6) and (7), as follows:(10)W=w+Δw
where *w* is the fixed length, determined by the actual data. Δw is the dynamic length, defined by the motion characteristics, and the calculation method is as follows:(11)$$\Delta w=\left(\lambda \times R_{V}+\mu \times R_{E}\right)\times \frac{w}{2}$$
(12)$$\lambda =\left\{\begin{array}{c} 0\;0\leq \left|R_{V}\right|<\delta \;\\ 1\;\delta \leq \left|R_{V}\right| \end{array}\right.$$
(13)$$\mu =\left\{\begin{array}{c} 0\;0\leq \left|R_{E}\right|<\varepsilon \;\\ 1\;\varepsilon \leq \left|R_{E}\right| \end{array}\right.$$
where λ and μ are the selection factors. 0≤δ≤1 and 0≤ε≤1 are the regulation thresholds.

After segmenting the trajectory data through the dynamic split window, the trajectory data $SW_{k}$ at the *k*-th moment is reconstructed according to Equation (Equation 4). Thereby, a small sample chaotic time series prediction set is constructed as follows:(14)$$SCP_{k}=\left[X\left(1\right),X\left(2\right),\cdots ,X\left(l\right),\cdots ,X\left(L\right)\right]\;L=W-\left(m-1\right)\tau$$
where X(l) = [$x\left(l+\left(m-1\right)\tau \right)$,⋯, $x\left(l\right)$,$x^{2}\left(l+\left(m-1\right)\tau \right)$, ⋯, $x^{2}\left(l\right)$, ⋯, $x^{p}\left(l+\left(m-1\right)\tau \right)$, ⋯, $x^{p}\left(l\right)$]${}^{T}$. *L* is the number of phase points. X(l) is a *D*-dimensional phase points, $D=\sum_{k=0}^{p}C_{k+m-1}^{k}$. $x\left(l\right)\in SW_{k}$x(l) is the trajectory data. *m* is the embedding dimensions. τ is the delay time. *p* is the order of Volterra prediction model.

### 3.3. Prediction Model Construction

Since the dynamic split window focuses on the latest trajectory data, the small sample chaotic time series prediction set we constructed is optimized in terms of data breadth; however, the spatiotemporal features of the trajectories themselves cannot be ignored. To further explore the motion patterns of maneuvering targets, we improve the sensitivity of the algorithm to identify key points by extracting the spatiotemporal features of trajectory data, which is optimized in terms of data depth—see Figure 4.

The movement trajectory of a maneuvering target may seem haphazard, but there are precursors before the maneuver occurs. In order to visualize the change pattern of target maneuvers, the trajectory data are segregated by significant points. The spatiotemporal features of the trajectory are extracted as follows:(15)xj−xixj+xixj+xi22≥ρj−i≥σ
where $x\left(i\right)$ and $x\left(j\right)$ are the trajectory data, 1≤i<j. σ and ρ are the distance threshold and the position threshold, respectively.

The significant point in the dynamic split window $SW_{k}$ of the *k*-th moment is measured by Equation (Equation 15), which is recorded as $SP_{k}$:(16)$$SP_{k}=\left[x\left(sp_{1}\right),\cdots ,x\left(sp_{q}\right),\cdots ,x\left(sp_{Q}\right)\right]\;sp_{q}\in \left\{1,2,\cdots ,W\right\}$$
where $x\left(sp_{q}\right)$ is the *q*-th significant point, $sp_{q}$ is the position in $SW_{k}$. *W* is the length of dynamic split window.

The significant points spatially represent the different trends of the trajectories, while the trajectory data follow a temporal order. Therefore, combined with the characteristics of sigmoid function, the spatiotemporal features $\partial_{l}$ of of the trajectory data are mined:(17)∂l=alalQ+1Q+1k≤l<sp12×al2×alQ+1Q+1sp1≤l<sp2⋮q+1×alq+1×alQ+1Q+1spq≤l<spq+1⋮alspQ≤l≤k+W−1
where
al=1/11+e−10×l+51+e−10×l+5l∈k,k+W−1

The PSO model identification algorithm is a problem-oriented metaheuristic that gradually approximates the optimal solution through the update iteration of particle swarm. The fitness function is the central part of PSO model identification algorithm, which controls the direction of population optimization. For the particle ${\hat{H}}_{i}$, the fitness function of the original algorithm as follows:(18)$$\begin{array}{cc} f_{e}\left(e_{i}\right) & =\frac{1}{L}\sum_{l=1}^{L}{\left(x\left(l\right)-\hat{x}\left(l\right)\right)}^{2}\\ \; & =\frac{1}{L}\sum_{l=1}^{L}{\left(x\left(l\right)-{\hat{H}}_{i}\times X\left(l\right)\right)}^{2} \end{array}$$
where ${\hat{H}}_{i}$ is the *i*-th particle. *L* is the number of phase points in the small sample chaotic time series prediction set. x(l) is the trajectory data and $x\left(l\right)\in SW_{k}$. $X\left(l\right)$ is a D-dimensional phase points.

The velocity and position evolution rules of the *i*-th particle ${\hat{H}}_{i}$ are as follows:(19)$$v_{i}\left(t+1\right)=\omega \times v_{i}\left(t\right)+c_{1}\times r_{1}\times \left(P_{i}\left(t\right)-{\hat{H}}_{i}\left(t\right)\right)+c_{2}\times r_{2}\times \left(P_{g}\left(t\right)-{\hat{H}}_{i}\left(t\right)\right)$$
(20)$${\hat{H}}_{i}\left(t+1\right)={\hat{H}}_{i}\left(t\right)+v_{i}\left(t+1\right)$$
where *t* is the iteration number of the PSO algorithm. $v_{i}$ is the velocity. ω is the inertia weight. $c_{1}$ and $c_{2}$ are learning factors. $r_{1}$ and $r_{2}$ are random numbers between the interval [0, 1]. $P_{i}\left(t\right)$ is the individual historical optimal position. $P_{g}\left(t\right)$ is the overall historical optimal position.

In this section, we design an improved fitness function by incorporating spatiotemporal features into a small sample chaotic temporal prediction set. By improving the sensitivity of the algorithm to identify key points in the trajectory, the population is led to evolve toward the optimal solution. The improved fitness function is as follows:(21)$$f_{e}\left(e_{i}\right)=\frac{1}{L}\sum_{l=1}^{L}\left[\beta_{l}\times {\left(x\left(l\right)-{\hat{H}}_{i}\times X\left(l\right)\right)}^{2}\right]$$
(22)βl=bl/bl∑i=1Lbi∑i=1Mbil=1,2,⋯,L
where
$$b_{l}=\partial_{l+\left(m-1\right)\tau }+\cdots +\partial_{l}+\partial_{l+\left(m-1\right)\tau }^{2}+\cdots +\partial_{l}^{2}+\cdots +\partial_{l+\left(m-1\right)\tau }^{p}+\cdots +\partial_{l}^{p}$$
$\beta_{l}$ is the normalized effect factor, and $\partial_{l}$ is the trajectory spatiotemporal feature.

The PSO algorithm finds the global optimal position with the smallest fitness value, which is the optimal parameter value of Volterra prediction model. The first model identification applies the initial population strategy proposed in Section 4. As the dynamic split window is updated online, the saved optimal values are utilized to initialize the particle swarm to decrease the identification time. Ultimately, the identified model enables online trajectory prediction of the maneuvering target.

### 3.4. Trajectory Estimation Optimization

Since the trajectory segmentation method is adopted to accomplish online identification, there may be superposition of prediction errors. Thereby, leading to the degradation in prediction accuracy of the algorithm. To improve the prediction performance, we propose a feedback optimization strategy of residual compensation to correct the predicted values. The feedback optimization strategy of residual compensation is shown in Figure 5. The calculation method is as follows:(23)$${\hat{x}}_{adj}\left(k+1\right)=\hat{x}\left(k+1\right)+resi\left(k+1\right)$$
(24)$$resi\left(k+1\right)=SW_{k}\times \theta$$
where ${\hat{x}}_{adj}\left(k+1\right)$ and $\hat{x}\left(k+1\right)$ represent the corrected prediction and model prediction, respectively. resi(k+1) is the residual compensation. $SW_{k}$ is the dynamic split window. θ is the compensation coefficient vector.

The least square (LS) algorithm is a simple linear recursive estimator with the advantages of small memory and fast calculation speed. In this paper, the LS algorithm is used to solve the compensation coefficient vector θ. The input item is $SCP_{k}$, and the expected output is ΔE:(25)$$\Delta E=\left[e(k-W+1),e(k-W+2),\cdots ,e(k)\right]$$
where
$$e\left(k\right)=x\left(k\right)-\hat{x}\left(k\right)$$
e(k) is the error between the true value and the predicted value. ΔE is the error vector of $SW_{k}$. *W* is the dynamic split window length.

## 4. Initial Population Strategy for PSO Model Identification Algorithm

The PSO model identification algorithm is an intelligent optimization algorithm with simpler structure, easier convergence and stronger robustness. However, the method suffers from the initial value sensitivity problem, and it is prone to fall into the local optimum trap in the process of seeking the optimal. To avoid this problem, on the premise of ensuring randomness, we propose a new initial population strategy to improve the optimization performance of the identification algorithm.

### 4.1. Chaotic Population Initialization

Since chaotic motion has obvious characteristics of nonlinearity, randomness and ergodicity, initializing the particle swarm with the idea of chaos can improve the quality of the population. The chaotic initialization method utilizes chaotic mapping and obtains the initial population through iterative means; however, it is difficult to traverse the entire search space when the number of iterations is insufficient. Reverse learning is an advancement of the traditional iterative approach, the idea of which is considering the opposite direction when generating a feasible solution. The schematic diagram is shown in Figure 6. Thus, it is combined with the reverse learning method to generate chaotic initial populations.

A tent map is a chaotic map in its parameter range, with uniform distribution function and good correlation. The tent chaotic map model is as follows:(26)Tk+1=TkTkννTk∈[0,ν)1−Tk1−Tk1−ν1−νTk∈[ν,1]
where ν∈(0,1) is the model parameter. When $T_{k}\in \left[0,1\right]$ and $T_{k}\neq \nu$, the above mapping is in a chaotic state.

We assume that the population size is *N*, and the search space is *D*-dimensional. The upper and lower search limits for each dimension are $x_{\min}^{d}$ and $x_{\max}^{d}$, respectively.

Firstly, it generates a *D*-dimensional random vector $T_{1}$:(27)$$T_{1}=\left[t_{1}^{1},t_{1}^{2},\cdots ,t_{1}^{d},\cdots ,t_{1}^{D}\right]\;t_{1}^{d}\in \left(0,1\right)$$

Secondly, each dimension of $T_{1}$ is substituted into Equation (Equation 26) for iteration. After N/2 iterations, a group of *D*-dimensional chaotic vectors are generated:(28)$$T_{n}=\left[t_{n}^{1},t_{n}^{2},\cdots ,t_{n}^{d},\cdots ,t_{n}^{D}\right]\;n=1,2,\cdots ,N/2$$
where the value of each vector $t_{n}^{d}\in \left(0,1\right)$, and d=1,2,⋯,D is the dimension of the search space.

Then, all chaotic vectors are mapped to the search space to generate the candidate population set $C_{1}=\left\{CX_{n}\right\}$:(29)$$CX_{n}=\left[cx_{n}^{1},cx_{n}^{2},\cdots ,cx_{n}^{d},\cdots ,cx_{n}^{D}\right]$$
where
$$cx_{n}^{d}=\left(x_{\max}^{d}-x_{\min}^{d}\right)\times t_{n}^{d}+x_{\min}^{d}$$

Finally, the reverse learning method is applied to generate the candidate population set $C_{2}=\left\{RX_{n}\right\}$:(30)$$RX_{n}=\left[rx_{n}^{1},rx_{n}^{2},\cdots ,rx_{n}^{d},\cdots ,rx_{n}^{D}\right]$$
where
$$rx_{n}^{d}=x_{\max}^{d}-x_{\min}^{d}+cx_{n}^{d}$$

The chaotic candidate population set $\left\{C_{1}\cup C_{2}\right\}$ generated by the above method take into account both diversity and ergodicity. However, due to the randomness of the chaotic mapping, the distribution of candidate populations in the search space is not uniform. Thus, the algorithm is prone to fall into the local optimum trap.

### 4.2. Spatial Distribution Compensation

To ensure the homogeneity of the initial population, we propose a spatial distribution compensation method to optimize the chaotic initialization, which makes the diversity and homogeneity of the initial population guaranteed.

All dimensions of the search space are divided equally, and interval values $I_{k}^{d}$ and spatial interval $S_{j}^{d}$ are generated in each dimension, as follows:(31)Ikd=xmind+k−1×xmaxd−xmind/xmaxd−xmindQQk=1,2,⋯,Q+1
(32)$$S_{j}^{d}=\left[I_{j}^{d},I_{j+1}^{d}\right]\;j=1,2,\cdots ,Q$$
where *Q* is the average number of divisions. d=1,2,⋯,D is the dimension of the search space.

Spatial interval $S_{j}^{d}$ divides each dimension of the search space equally into discrete grid states, as shown in Figure 7, where each grid represents an interval.

From the distribution of the chaotic candidate population set in the search space, defining the following sparsity function $f_{s}\left(S_{j}^{d}\right)$:(33)fsSjd=1−rjd×cjdcjdNN
where $r_{j}^{d}\in \left[0,1\right]$ is a randomly generated value. $c_{j}^{d}$ is the number of the candidate population in the *j*-th interval of *d*-th dimension. *N* is the population size.

The larger the sparsity value, the fewer candidate populations fall in the interval; therefore, calculate the sparsity values of each interval and arrange in descending order, and select the median of the *M* intervals with the greatest sparsity as the new candidate populations.
(34)sxid=Ii+1d−Iid/Ii+1d−Iid22i=1,2,⋯,M
where *M*(M<Q) is determined by the actual distribution. $I_{i}^{d}$ is the *i*-th interval value after sorting according to sparsity.

Combining the interval median values calculated by Equation (Equation 34), it generates the candidate population position set $C_{3}=\left\{SX_{i}\right\}$:(35)$$SX_{i}=\left[sx_{i}^{1},sx_{i}^{2},\cdots ,sx_{i}^{d},\cdots ,sx_{i}^{D}\right]$$

In summary, there are generated $\left(N+M\right)$ groups of candidate population set C:(36)$$C=\left\{C_{1}\cup C_{2}\cup C_{2}\right\}=\left\{CX_{n}\cup RX_{n}\cup SX_{i}\right\}\;n=1,2,\cdots ,N/2\;i=1,2,\cdots ,M$$

We select *N* groups of candidate populations with the best fitness as the initial populations of the PSO model identification algorithm. The proposed initial population strategy avoids the algorithm from falling into local traps by increasing the diversity and homogeneity of the initial population. Meanwhile, it improves the efficiency of the algorithm in finding the optimum.

## 5. Algorithm Implementation

For the online prediction scenario of maneuvering target trajectory, the OTP-SSCT algorithm is proposed in this paper. Firstly, we initialize the algorithm parameters and use the proposed initial population strategy to obtain the initial population of the PSO model identification algorithm. Secondly, the split window length is dynamically adjusted by the motion characteristics of the maneuvering target, and a small sample chaotic time series prediction set is constructed. Then, we use the improved PSO algorithm to identify the trajectory prediction model. Finally, the prediction results are obtained, while the feedback optimization strategy is applied to further improve the prediction accuracy. According to the above analysis, more details related to the proposed OTP-SSCT algorithm, and the pseudo-code is shown in Algorithm 1.
**Algorithm 1:** OTP-SSCT algorithm.**Input:** Trajectory data $x\left(n\right),n=1,2,\cdots ,N$**Output:** Prediction results $\hat{x}\left(n\right),n=W,W+1,\cdots ,N,\cdots$
1:Initializing the dynamic split window length *W*, embedding dimension *m* and delay time τ based on the historical trajectory data;2:Obtaining the initial population according to (26)–(36);3:**for** k=1:N **do**4:  Calculating the similarity distance DTW according to (5);5:  **if** $DTW>TH_{dtw}$ **then**6:   Adjusting the dynamic split window length *W* according to (10)–(13);7:  **end if**8:  Segmenting the trajectory data x(n) according to the dynamic split window;9:  Constructing the small sample chaotic time series prediction set $SCP_{k}$ according to (14);10:  Extracting trajectories spatiotemporal features $\partial_{n}$ according to (17);11:  Identifying the model parameters ***H*** by the improved PSO model identification algorithm according to (19)–(22);12:  Constructing the trajectory prediction model according to (4);13:  Obtaining the prediction results $\hat{x}\left(n\right)$ and performing the feedback optimization strategy according to (23)–(25);14:**end for**

## 6. Simulation and Analysis

To address the problem of stable online prediction of maneuvering target trajectories, this paper proposes the OTP-SSCT algorithm. Considering the motion characteristics of trajectory data, the improved PSO model identification algorithm is used to identify the parameters of Volterra model. Finally, the online trajectory prediction of maneuvering target is realized.

In this experiment, considering the maneuvering characteristics of the UAV trajectory, we choose the 3rd-order Volterra prediction model as the structural basis of the identification model, as follows:(37)$$\begin{array}{cc} \hat{x}\left(n+1\right) & =\sum_{k=1}^{3}\sum_{i_{1},i_{2},i_{3}=0}^{m-1}h_{k}\left(i_{1},i_{2},i_{3}\right)\prod_{j=1}^{k}x\left(n-i_{j}\times \tau \right)\\ \; & =\sum_{i_{1}=0}^{m-1}h_{1}\left(i_{1}\right)x\left(n-i_{1}\times \tau \right)+\sum_{i_{1}=0}^{m-1}\sum_{i_{2}=i_{1}}^{m-1}h_{2}\left(i_{1},i_{2}\right)x\left(n-i_{1}\times \tau \right)x\left(n-i_{2}\times \tau \right)\\ \; & +\sum_{i_{1}=0}^{m-1}\sum_{i_{2}=i_{1}}^{m-1}\sum_{i_{3}=i_{2}}^{m-1}h_{3}\left(i_{1},i_{2},i_{3}\right)x\left(n-i_{1}\times \tau \right)x\left(n-i_{2}\times \tau \right)x\left(n-i_{3}\times \tau \right) \end{array}$$
where $\hat{x}\left(n+1\right)$ and x(n) represent the prediction result and the trajectory data, respectively. *m* and τ represent the memory length of the model and the delay time, respectively. $h_{k}\left(i_{1},i_{2},i_{3}\right)$ is the kernel parameter.

To verify the performance of our OTP-SSCT algorithm, the following simulation experiments are designed in three directions.

(1) For the initial population setting problem of the PSO algorithm, different population selection strategies are used for particle swarm initialization and model parameter identification. The performance of the proposed population initialization strategy is verified by comparing the identification accuracy.

(2) The model identification of chaotic time series in Lorenz-X dimension is performed with different optimization algorithms. In comparison with several performance metrics, it tests the identification capability of the proposed improved PSO model identification algorithm.

(3) Aiming at online trajectory prediction scenarios of maneuvering targets, different prediction algorithms are adopted for online trajectory prediction of UAV three-degree-of-freedom maneuvering target. By analyzing the precision metrics such as prediction accuracy and response time, it verifies the performance of the proposed OTP-SSCT algorithm.

This paper uses four precision metrics for the quantitative evaluation: mean square error (MES), root mean square error (RMSE), mean absolute percentage error (MAPE) and coefficient of determination $R^{2}$. The software used for the simulation experiment is MATLAB2019a. The simulation environment: CPU is Intel Core i5-6500 3.20 GHz, memory is 8GB, and the operating system is Microsoft Windows 10.

### 6.1. Parameter Identification Experiment with Different Initialization Methods

Initial population directly affects the optimization performance of PSO algorithm; therefore, given the real parameters, the performance of our proposed initialization strategy is tested by comparing the identification errors of different initialization methods. The experimental process is shown in Figure 8.

The comparison methods are random initialization method, chaotic initialization method and sequence initialization method [28]. Identifying a 3rd-order Volterra model with a memory length of 3. The given kernel parameter vector is as follows:$$\begin{array}{c} H=[1.54,-1,0.56,0.87,0,0,1.13,1.21,-1.29,1.45,0,0,0.88,0,0,-0.66,0,0,1.06{]}^{T} \end{array}$$

For the Volterra model with given kernel parameters ***H***, the input signal $x\left(k\right)\in \left[0,1\right]$ is uniform white noise, and the output signal is $y\left(k\right)$. $x\left(k\right)$ and $y\left(k\right)$ constitute the test set, and the PSO algorithm with four initialization methods is used for parameter identification, respectively. Setting the population size to 100 and the number of iterations to 500. Each method is simulated for 50 times, and the identification results are averaged.

It can be seen from Figure 9 and Table 1 that the PSO algorithm using four different initialization methods can identify approximate values close to the true values of the kernel parameters. By comparing the absolute error between the identification results and the true values, the minimum error of our method is about 0.0019, the maximum error is about 0.0143, and the average error is about 0.0053. It can be seen that the average error of our method is 73.3%, 65.6% and 36.9% smaller than that of the random initialization method, chaotic initialization method and sequence initialization method, respectively; therefore, the initial population strategy proposed in this paper significantly improves the identification effect of the PSO model identification algorithm.

### 6.2. Model Identification Experiment with Different Optimization Algorithms

To further evaluate the performance of our algorithm in identifying the model parameters. Firstly, the X-dimensional time series generated by the Lorenz equation is used as validation data, and it is divided into historical training set and online prediction set. Secondly, the algorithm parameters are determined and the training set is used to identify the model. Finally, the prediction set is sampled by the dynamic split window, identifies the prediction model online and performs single-step prediction. In addition, improved particle swarm optimization algorithm (IPSO) [29], genetic algorithm (GA) [30] and beetle antennae search algorithm (BAS) [31] are used for comparative analysis. The parameter of the algorithms are shown in Table 2.

The Lorenz equation is a classical chaotic system, and the generated chaotic time series are often used for algorithm verification. The mathematical expression of the Lorenz equation is as follows:$$\left\{\begin{array}{cc} x^{\prime }= & -\sigma (x-y)\\ y^{\prime }= & (r-z)x-y\\ z^{\prime }= & xy-bz \end{array}\right.$$

Based on the above Lorenz equation, we set the initial value to (1,0,0), σ=10, r=28 and $b=\frac{8}{3}$. The fourth-order Runge–Kutta method is used for sampling, with an integration step of 0.02. In the generated chaotic time series, 700 sample points of the Lorenz-X dimensional are selected as the experimental data. The initial length of dynamic split window is set to 200, and the last 500 sample points are used as the test set. The delay time τ=8 and the embedding dimension m=5 are determined by the historical trajectory. To ensure the objectivity of comparison experiment, the population size of each algorithm is 100 and the maximum number of iterations is 500.

Figure 10 shows the trajectory prediction of Lorenz-X dimensional by four model identification algorithms. It can be seen that the models identified by the four algorithms can achieve nonlinear trajectory prediction, and the predicted trajectory of our algorithm has best fit with the true trajectory, while the BAS algorithm has the most deviation points. Figure 11 shows the trajectory prediction errors of the four algorithms, where $Err=\left|x\left(n\right)-\hat{x}\left(n\right)\right|$. It can be found that our algorithm has only 7.2% of prediction errors greater than 0.2, compared to 21.4%, 42% and 38.4% for IPSO, GA and BAS algorithms, respectively. Therefore, the identification model of our algorithm can control the prediction error within a low fluctuation range, which indicates that the model identification performance is much higher than the other three algorithms.

From Figure 12, the average prediction error of our algorithm are only 35.9%, 16.2%, and 13.6% of IPSO, GA and BAS algorithms, respectively, and the maximum prediction error are only 50.1%, 38.9% and 24.5%, respectively. Obviously, the prediction accuracy of the model identified by our algorithm is higher than the other three optimization algorithms. According to the maneuver of the Lorenz-X dimensional trajectory, it is calculated the average prediction error at each 50 steps, as shown in Figure 13. It is shown that our algorithm can maintain a high prediction accuracy for strongly nonlinear trajectory over time, while the other three algorithms are affected by the fluctuations of trajectories. Meanwhile, it indicates that the proposed algorithm is highly robust and can achieve model identification of strongly nonlinear trajectories.

Table 3 shows that our algorithm has the smallest values of both MSE and RMSE, and $R^{2}$ is closest to 1. It shows that our algorithm has a significant advantage in nonlinear trajectory prediction. Compared with the other three algorithms, the MAPE value of our algorithm is the lowest, indicating that the OTP-SSCT algorithm has the best stability. The statistics in Table 3 show the average results from 50 independent replicate experiments for each algorithm. In summary, the proposed algorithm has certain advantages in terms of prediction accuracy and stability.

### 6.3. Online Trajectory Prediction Experiment for Maneuvering Targets

In order to test the performance of the OTP-SSCT algorithm for the online prediction of maneuvering target trajectories, the mapping *f* in Equation (Equation 2) is replaced by LSTM algorithm, kernel least mean squares (KLMS) algorithm, kernel recursive least squares (KRLS) algorithm and support vector regression (SVR) algorithm. We perform online single-step prediction of maneuvering target trajectories using the five algorithms separately, and a comparative analysis is given by precision metrics.

The trajectory maneuver can be divided into horizontal maneuvers, vertical maneuvers and spatial combination maneuvers. We adopt the UAV three-degree-of-freedom model to simulate the generation of real-time trajectory data. Considering the UAV as a mass point and the ground coordinate system as an inertial coordinate system, the motion equations are as follows:(38)x˙t=vtcosθtcosψty˙t=vtcosθtsinψtz˙t=vtcosθtv˙t=gFt−sinθtθ˙t=gvtTtcosϕt−cosθtψ˙t=gTtsinϕtgTtsinϕtvtcosθtvtcosθt
where $x_{t}$, $y_{t}$ and $z_{t}$ represent the horizontal and height coordinates of the UAV in the inertial coordinate system, respectively. $v_{t}$, $\theta_{t}$ and $\psi_{t}$ represent the speed, climb angle and heading angle of the UAV, respectively. $\phi_{t}$ is the roll angle. *g* is the gravitational acceleration. $F_{t}$ and $T_{t}$ represent the horizontal overload and the longitudinal overload, respectively. ${\left[x_{t},y_{t},z_{t},v_{t},\theta_{t},\psi_{t}\right]}^{T}$ and ${\left[F_{t},T_{t},\phi_{t}\right]}^{T}$ are the state variables and control variables of the UAV, respectively.

A randomly generated maneuvering trajectory by Equation (Equation 38) was used as the experimental data. The number of sensor samples is 300 and the sampling interval is 0.2 s. The initial length of dynamic split window is set to 20, and predict the latter 280 trajectory data online. Adopt three-dimensional coordinate independent prediction to improve the prediction accuracy. The delay time τ and the embedding dimension *m* of each dimension of the trajectory data are determined by the historical trajectory. X-dimension: τ=5, m=2. Y-dimension: τ=6, m=2. Z-dimension: τ=6, m=2.

As can be seen from Figure 14, the five prediction trajectories roughly match the true trajectories. Among them, our algorithm is closest to the true trajectory, while the other four algorithms all appear a lot of deviation points. Figure 15 shows the online trajectory prediction errors for each dimension. From Figure 15, the farthest distance in space between the predicted trajectory and the true trajectory can be calculated, where ours, LSTM, KLMS, KRLS and SVR algorithms are about 1.34 m, 8.57 m, 9.54 m, 2.55 m and 8.93 m, respectively. Meanwhile, the average prediction error of our algorithm improves approximately 87.6%, 87.3%, 45.9% and 78.5% over LSTM, KLMS, KRLS and SVR algorithms, respectively. Obviously, our algorithm has the smallest error fluctuation range and the strongest robustness performance. The mean absolute error is calculated for every 20 steps, as shown in Figure 16. It can be found that the prediction accuracy of our algorithm for maneuvering target trajectory is better than the other four algorithms, which can achieve high accuracy online trajectory prediction.

From the precision metrics in Table 4, it can be seen that our algorithm has optimal trajectory prediction results for the maneuvering target and it can accurately track the true trajectory in the whole stage. The LSTM algorithm uses gradient descent algorithm to learn the long-term dependencies of data through three gating units. The network parameters are trained by the dynamic split window in this experiment. Since the algorithm is prone to falling into local extremes or overfitting, and the insufficient amount of training data can also lead to model training failure, which affects the prediction performance. The KLMS and KRLS algorithms are adaptive filtering algorithms based on kernel method. In this experiment, the Gaussian kernel function is chosen as the kernel function. Both algorithms require only a small amount of computation for filtering estimation and weight update, but memory consumption will increase linearly with the number of iterations. Although there are some sparse methods such as novel criterion and approximate linear correlation, they still cannot maintain the prediction accuracy of the algorithm. The SVR algorithm is used to find a minimum interval band to wrap all the training samples. In this experiment, we use the dynamic split window to segment the training and prediction sets. It can be found that the SVR algorithm has a large prediction error at maneuvering turns, which is due to the poor preprocessing of the training data thus leading to parameter identification failure.

Figure 17 shows the single-step predicted response time of the five algorithms for 50 independent replicate experiments, in which the average single-step prediction time of our algorithm, LSTM, KLMS, KRLS and SVR algorithms are about 0.03352, 0.04344, 0.03257, 0.03604 and 0.04102, respectively. Among them, the single-step prediction time of our algorithm is about 89.8% of five algorithms on average, indicating that our algorithm has an outstanding position in terms of response time. Notice that the single-step prediction time of our algorithm is slightly longer than that of the KLMS algorithm. This is due to the first model identification adopting the improved initial population strategy, which consumes more time. Although the saved optimal values are later used for population initialization, maneuvering changes of the target still increase the identification time. However, it is undeniable that our algorithm outperforms the KLMS algorithm in terms of prediction accuracy.

## 7. Conclusions

In this paper, we propose the OTP-SSCT algorithm to achieve a stable online prediction of maneuvering target trajectories. The algorithm has the following advantages: (1) We build an online prediction model of maneuver trajectory according to chaotic time series analysis. Thus, a small sample chaotic time series prediction set that characterizes the target motion characteristics is constructed through the trajectory segmentation method. By fully considering the nonlinear maneuver characteristics of trajectory data, we introduce the spatiotemporal features into the PSO model identification algorithm, thus improving the identification sensitivity of key points. Meanwhile, the feedback optimization strategy is used to avoid the superposition of prediction errors and improve the prediction accuracy. (2) We propose a new initial population strategy to solve the initial value sensitivity problem of PSO model identification algorithm, thus improving the effectiveness of the initial parameters on model identification. In our future work, it will be interesting to further improve the speediness of the algorithm.

## Figures and Tables

**Figure 1 entropy-24-01668-f001:**
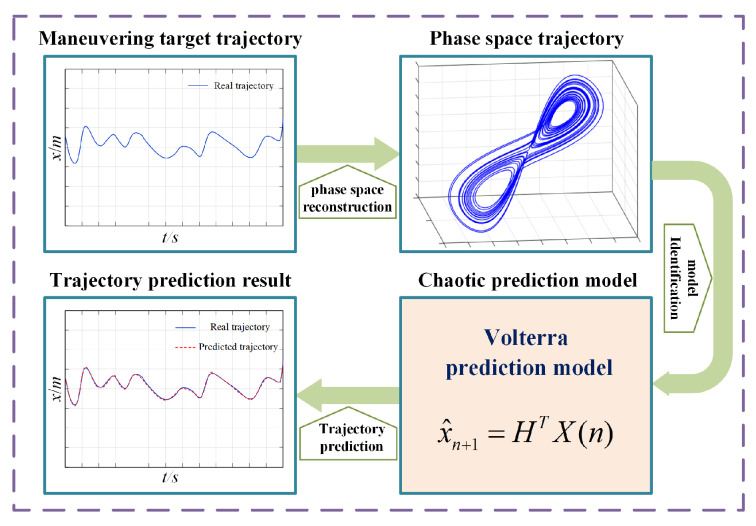
Trajectory prediction process of Volterra model.

**Figure 2 entropy-24-01668-f002:**
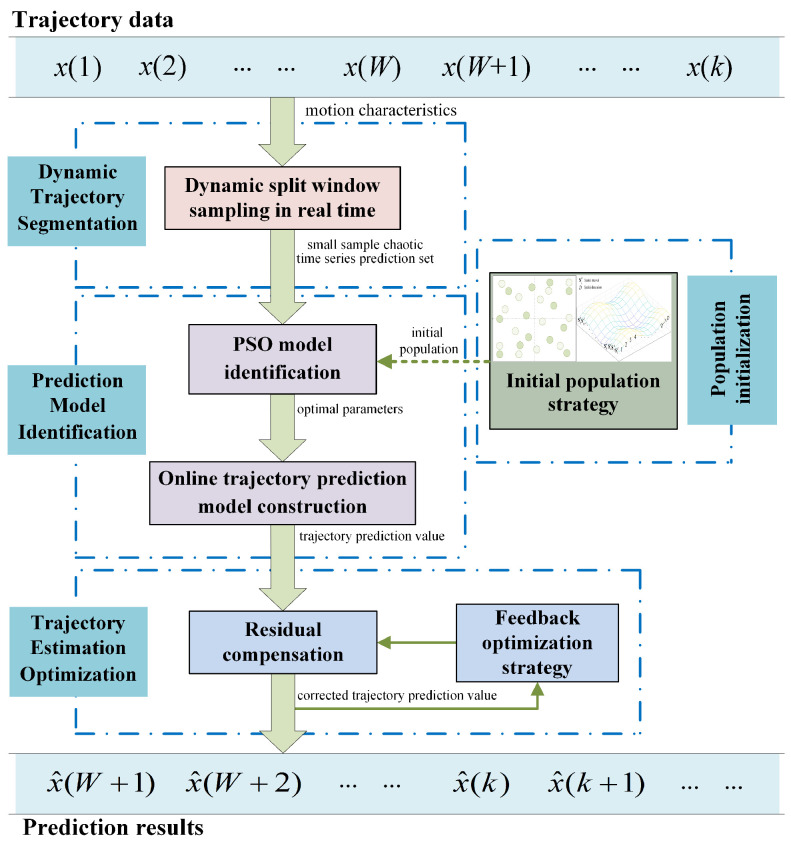
Framework diagram of the proposed algorithm.

**Figure 3 entropy-24-01668-f003:**
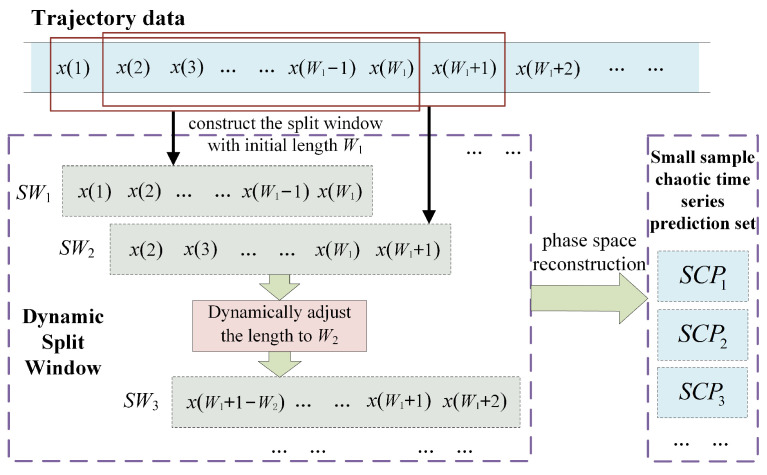
Flow chart of trajectory segmentation with dynamic split window.

**Figure 4 entropy-24-01668-f004:**
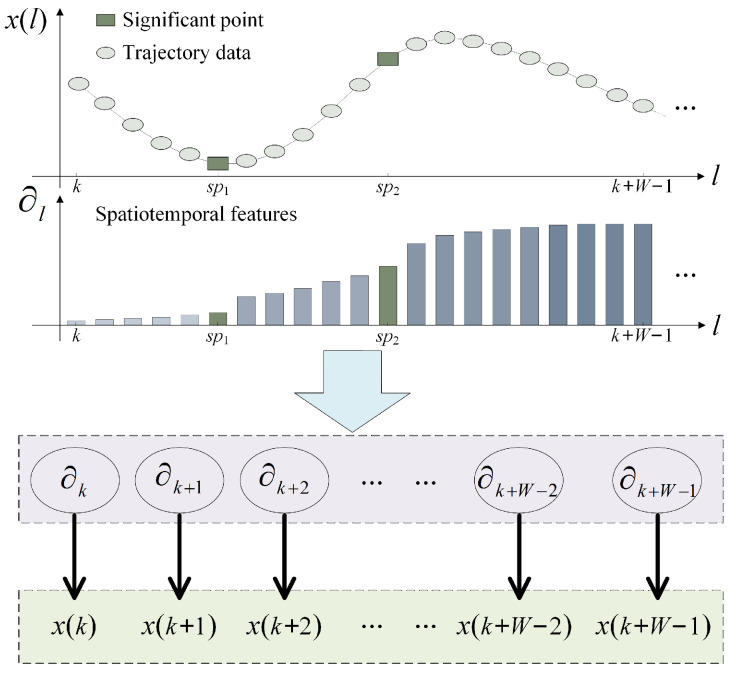
Extraction diagram of trajectory spatiotemporal features.

**Figure 5 entropy-24-01668-f005:**
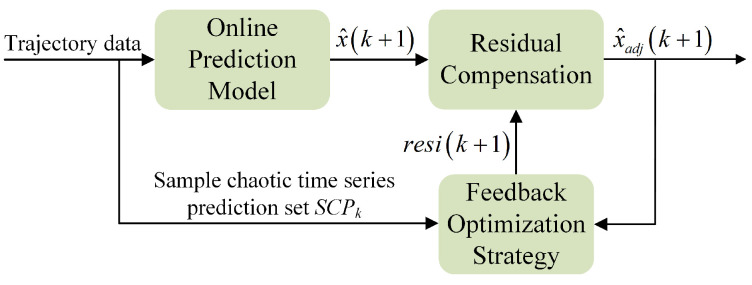
Feedback optimization strategy of residual compensation.

**Figure 6 entropy-24-01668-f006:**
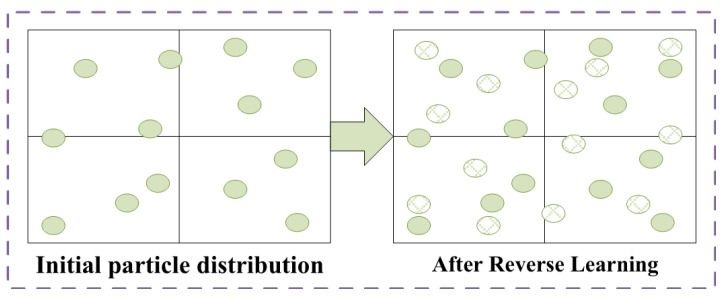
Reverse optimization of initial population.

**Figure 7 entropy-24-01668-f007:**
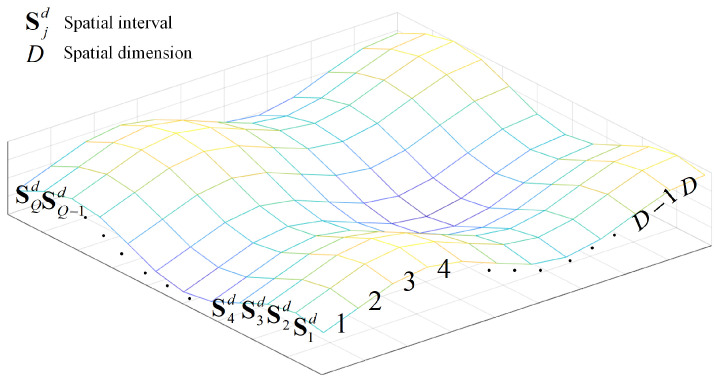
Discrete gridding of the search space.

**Figure 8 entropy-24-01668-f008:**
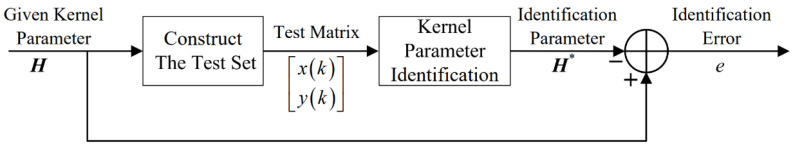
Flowchart of *experiment* 1.

**Figure 9 entropy-24-01668-f009:**
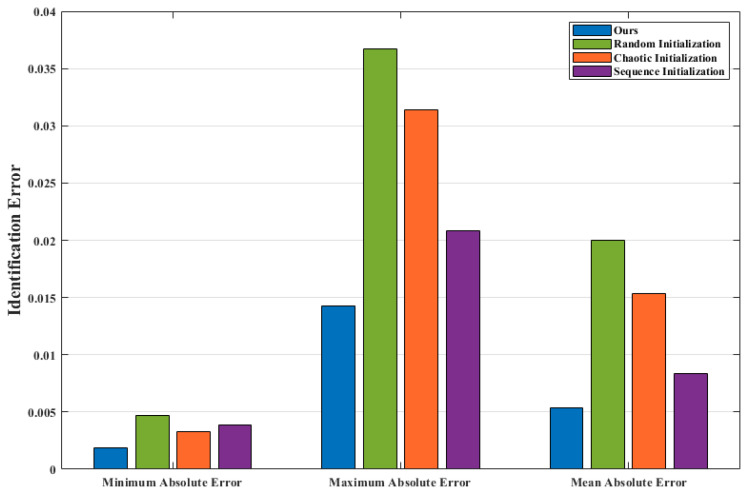
Comparison of parameter identification error.

**Figure 10 entropy-24-01668-f010:**
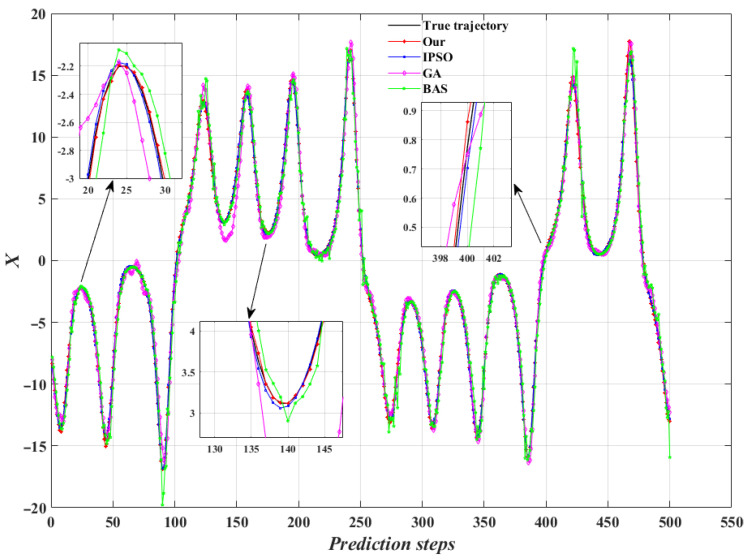
Trajectory prediction of Lorenz-X dimensional.

**Figure 11 entropy-24-01668-f011:**
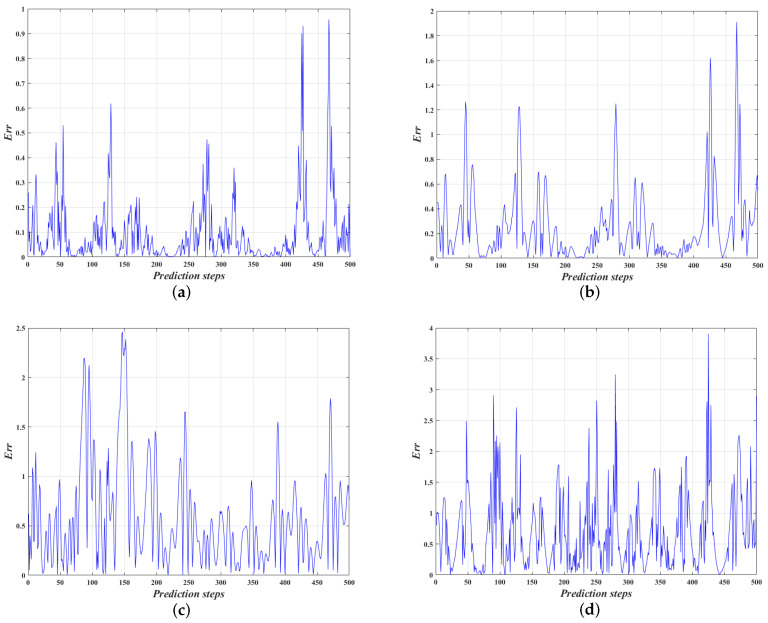
Trajectory prediction errors of the four algorithms. (**a**) Our (OTP-SSCT) algorithm; (**b**) the IPSO algorithm; (**c**) the GA algorithm; (**d**) the BAS algorithm.

**Figure 12 entropy-24-01668-f012:**
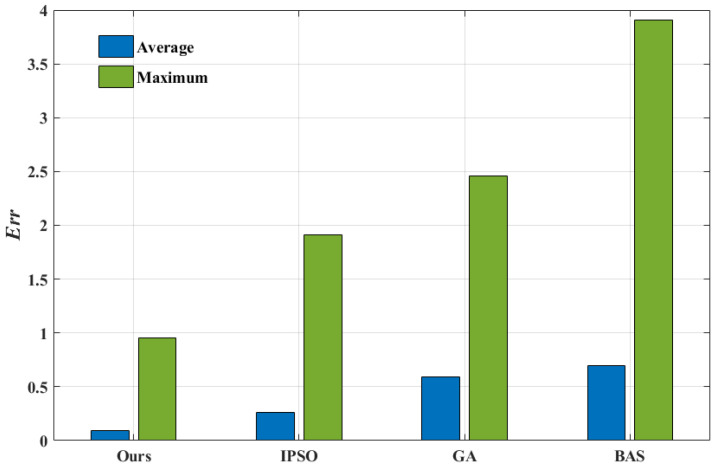
Average and maximum prediction errors of the four algorithms.

**Figure 13 entropy-24-01668-f013:**
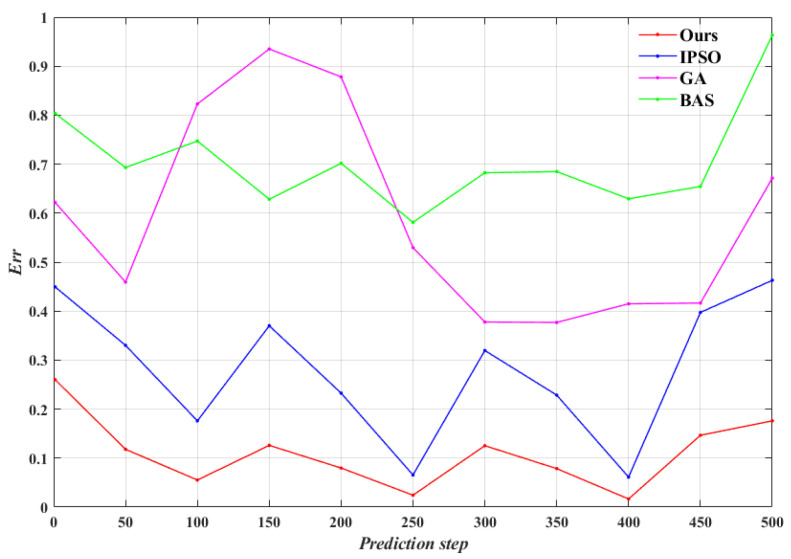
Average prediction error of the prediction process in *experiment* 2.

**Figure 14 entropy-24-01668-f014:**
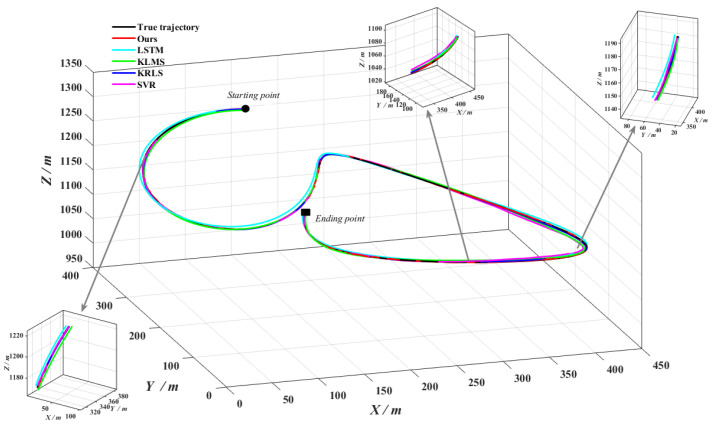
Online trajectory prediction of maneuvering target.

**Figure 15 entropy-24-01668-f015:**
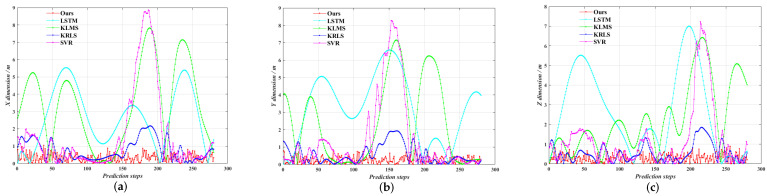
Online trajectory prediction errors of five algorithms. (**a**) X-dimension prediction errors; (**b**) Y-dimension prediction errors; (**c**) Z-dimension prediction errors.

**Figure 16 entropy-24-01668-f016:**
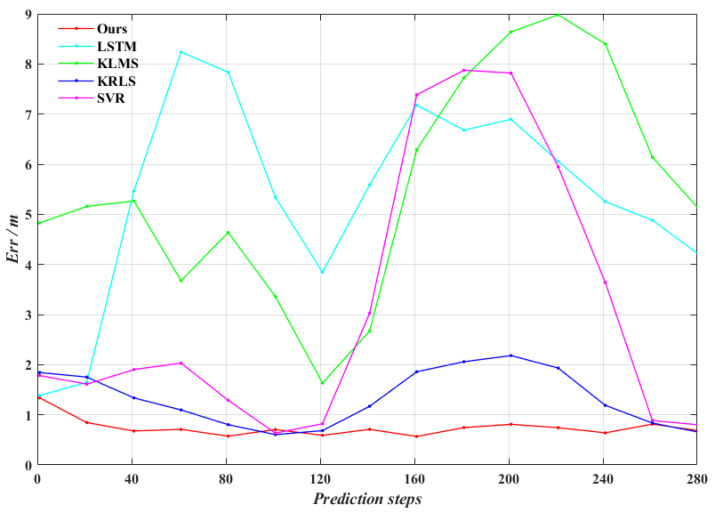
Average prediction error of the prediction process in *experiment* 3.

**Figure 17 entropy-24-01668-f017:**
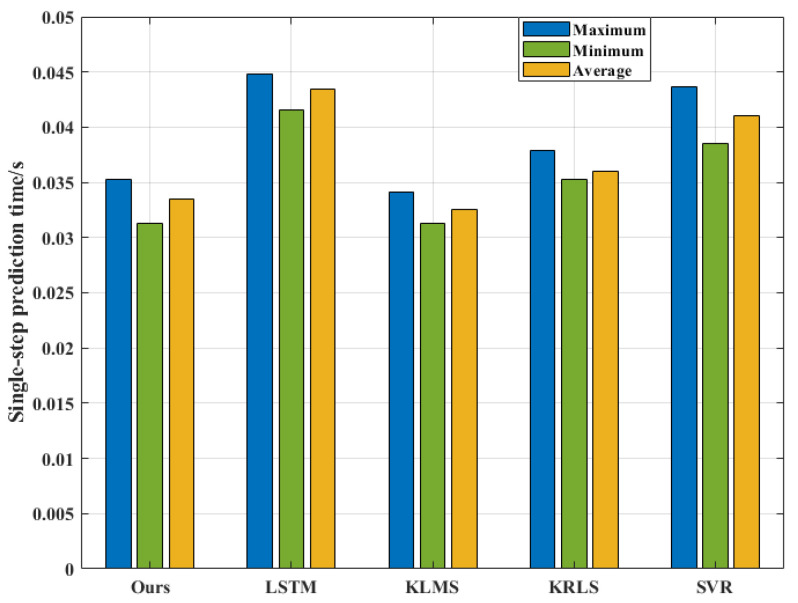
Comparison of single-step prediction times.

**Table 1 entropy-24-01668-t001:** Kernel parameter identification results.

*H* ^1^	T-V ^2^	Ours		R-I ^3^		C-I ^4^		S-I ^5^	
		*H** ^6^	*E* ^7^	*H**	*E*	*H**	*E*	*H**	*E*
$h_{1}\left(0\right)$	1.54	1.5374	0.0026	1.5096	0.0304	1.5367	0.0033	1.5354	0.0046
$h_{1}\left(1\right)$	−1	−0.9935	0.0065	−0.9633	0.0367	−0.9735	0.0265	−0.9792	0.0208
$h_{1}\left(2\right)$	0.56	0.5554	0.0046	0.5301	0.0299	0.5473	0.0127	0.5503	0.0097
$h_{2}\left(0,0\right)$	0.87	0.8738	0.0038	0.8752	0.0052	0.8734	0.0034	0.8750	0.0050
$h_{2}\left(0,1\right)$	0	$-5.6\times 10^{-4}$	$5.6\times 10^{-4}$	−0.0074	0.0074	−0.0098	0.0098	−0.0060	0.0060
$h_{2}\left(0,2\right)$	0	$-8.6\times 10^{-4}$	$8.6\times 10^{-4}$	0.0086	0.0086	0.0093	0.0093	0.0074	0.0074
$h_{2}\left(1,1\right)$	1.13	1.1377	0.0077	1.0990	0.0310	1.0986	0.0314	1.1193	0.0107
$h_{2}\left(1,2\right)$	1.21	1.2081	0.0019	1.2053	0.0047	1.2028	0.0072	1.2061	0.0039
$h_{2}\left(2,2\right)$	−1.29	−1.2833	0.0067	−1.2636	0.0264	−1.2718	0.0182	−1.2807	0.0093
$h_{3}\left(0,0,0\right)$	1. 45	1.4524	0.0024	1.4567	0.0067	1.4395	0.0105	1.4436	0.0064
$h_{3}\left(0,0,1\right)$	0	$-4.3\times 10^{-5}$	$4.3\times 10^{-5}$	−0.0336	0.0336	0.0078	0.0078	0.0160	0.0160
$h_{3}\left(0,0,2\right)$	0	$7.2\times 10^{-4}$	$7.2\times 10^{-4}$	0.0011	0.0011	0.0441	0.0441	0.0051	0.0051
$h_{3}\left(0,1,1\right)$	0.88	0.8850	0.0050	0.8971	0.0171	0.8554	0.0246	0.8862	0.0062
$h_{3}\left(0,1,2\right)$	0	−0.0079	0.0079	0.0255	0.0255	0.0028	0.0275	−0.0466	0.0466
$h_{3}\left(0,2,2\right)$	0	0.0045	0.0045	−0.0121	0.0121	0.0516	0.0516	0.0162	0.0162
$h_{3}\left(1,1,1\right)$	−0.66	−0.6569	0.0031	−0.6408	0.0192	−0.6353	0.0247	−0.6682	0.0082
$h_{3}\left(1,1,2\right)$	0	−0.0065	0.0065	−0.0910	0.0910	0.0402	0.0402	0.0202	0.0202
$h_{3}\left(1,2,2\right)$	0	0.0105	0.0105	0.0765	0.0765	−0.0085	0.0085	0.0040	0.0040
$h_{3}\left(2,2,2\right)$	1.06	1.0457	0.0143	1.0474	0.0126	1.0534	0.0066	1.0526	0.0074

Explain: ^1^ ***H*** represents the kernel parameters; ^2^ **T-V** represents true value; ^3^ **R-I** represents random initialization; ^4^ **C-I** represents chaotic initialization; ^5^ **S-I** represents sequence initialization; ^6^ ***H**** represents the identification result; ^7^ ***E*** represents the identification error.

**Table 2 entropy-24-01668-t002:** Algorithm parameter setting of *experiment* 2.

Algorithm	Parameter Settings
Ours	Q=29, w=0.8, $c_{1}=c_{2}=2$, $r_{1}=rand$, $r_{2}=rand$
IPSO	S=5, $\lambda_{1}=\lambda_{2}=\lambda_{3}=2$, $\alpha_{1}=rand$, $\alpha_{2}=rand$, $\alpha_{2}=rand$
GA	$p_{c}=0.8$, $p_{m}=0.01$
BAS	η=0.95, δ=2

**Table 3 entropy-24-01668-t003:** Comparison of precision metrics of *experiment* 2.

Trajectory	Algorithm	Precision Metrics			
		MSE	RMSE	MAPE	*R* ${}^{2}$
Lorenz-X	Ours	0.0262	0.1618	0.0235	0.9996
	IPSO	0.1575	0.3969	0.0702	0.9975
	GA	0.5872	0.7663	0.1844	0.9907
	BAS	0.8765	0.9362	0.2098	0.9861

**Table 4 entropy-24-01668-t004:** Comparison of precision metrics of *experiment* 3.

Prediction Dimension	Prediction Algorithm	Precision Metrics			
		MSE/m^2^	RMSE/m	MAPE	*R* ${}^{2}$
X	Ours	0.3521	0.5934	0.0059	0.9998
	LSTM	14.943	3.8655	0.0388	0.9994
	KLMS	26.065	5.1054	0.0422	0.9988
	KRLS	1.5008	1.2251	0.0111	0.9997
	SVR	12.407	3.5224	0.0146	0.9993
Y	Ours	0.2383	0.4881	0.0053	0.9998
	LSTM	21.746	4.6633	0.0563	0.9985
	KLMS	17.012	4.1245	0.0504	0.9989
	KRLS	0.9433	0.9712	0.0126	0.9996
	SVR	10.144	3.1851	0.0338	0.9993
Z	Ours	0.2848	0.5337	$4.7\times 10^{-4}$	0.9999
	LSTM	17.764	4.2148	0.0036	0.9987
	KLMS	12.886	3.5897	0.0032	0.9991
	KRLS	0.7959	0.8921	$7.6\times 10^{-4}$	0.9997
	SVR	6.3458	2.5191	0.0017	0.9995

## Data Availability

Not applicable.

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
