# Peer review of "Online Tracking of Maneuvering Target Trajectory Based on Chaotic Time Series Prediction"

_entropy, 2022, doi:10.3390/e24111668_

Round 1

Reviewer 1 Report

To solve the problem of trajectory prediction of maneuvering targets, the authors propose a Volterra prediction method and model based on the analysis of chaotic time series of trajectory segmentation. To increase the sensitivity of identifying key points of trajectory data, spatio-temporal characteristics are introduced into the particle swarm optimization (PSO) algorithm. The authors also presented a new strategy for the initial population of the PSO algorithm. The subject of the manuscript is not novel but interesting. The literature review is deep and complete. I have some comments, which I think will improve the readability of the manuscript.

[Major points]

[1] The results are presented in a clear way, but for completeness of the analysis of the experimental results, precision is shown, but recall or F-score is not shown.

[Minor points]

[1] p.13, s.375 It will be good to put a comma when listing.

[2] p.18, s.464 “…The statistics in Table 3 show the average running time…” the table number is incorrect, there is no such table at all.

Reviewer 2 Report

The paper introduced an algorithm for the online prediction of maneuvering target trajectories. Some advantages of the proposed algorithm are the avoidance of the superposition of prediction errors and improve the prediction accuracy as well as improving the effectiveness of the initial parameters on model identification. Details of simulations and analysis were reported. Some comments for improving the paper are as follows.

(1)   I agree with the authors that your model identification algorithm is an intelligent optimization algorithm with simpler structure, easier convergence and stronger robustness. But I am worry about the speediness of the algorithm. How about this vital feature of the proposed algorithm.

(2)   Authors selected N groups of candidate populations with the best fitness as the initial populations of the PSO model identification algorithm. I do not understand this point clearly. How does the authors select the correct number of N?

(3) Details of simulations and analysis were reported. However, the algorithm implementation should be enlarged. 
